# Construction of a Highly Selective Membrane Sensor for the Determination of Cobalt (II) Ions

**Sabry Khalil** [1,*] **and Mohamed El-Sharnouby** [2]

[1] Department of Food Nutrition Science, College of Science, Taif University, P. O. Box 1099, Taif 21944, Saudia Arabia

[2] Department of Biotechnology, College of Science, Taif University, P.O. Box 11099, Taif 21944, Saudi Arabia; m.sharnouby@Tu.edu.sa

\* Correspondence: Sabry@Tu.edu.sa

**Abstract:** A highly Co (II) liquid ion-selective electrode depending on the reaction of cobalt ions with the reagent 2-(5-Bromo-2-pyridylazo)-5-[N-n-propyl-N-(3-sulfopropyl) amino] aniline is successfully fabricated. The characteristic slope (56.66 mV), the linear range response from $3.4 \times 10^{-8}$ to $2.4 \times 10^{-2}$ molar, the detection limit ($2.7 \times 10^{-8}$) molar, the selectivity coefficient toward some metal ions, the time of response (10 s), lifetime (seven months), the pH effect on the sensor potential and the basic analytical parameters were studied. The sensor was used to estimate the concentration of cobalt ions in food products and pharmaceutical formulations. The obtained results of the developed sensor were statistically analyzed and compared with those of other different reported electrodes.

**Keywords:** ion-associate complexe; membrane sensor; cobalt estimation; food products; pharmaceutical samples

## 1. Introduction

Cobalt is used to treat anemia with pregnant women, because it stimulates the production of red blood cells. The cobalt daily intake is variable and may be 1 mg, almost will pass through the body UN adsorbed, except in the case of vitamin B12 [1].

However, too high concentrations of cobalt may damage human health. When we breathe in too high concentrations through the air, we experience lung effects, such as asthma and pneumonia. Therefore, it is very important to detect $Co^{+2}$ in pharmaceuticals and food products.

Applications of ion-selective membrane sensors (ISEs) have evolved to be used in many different fields [2–4].

Several techniques including spectrophotometry [5–10], electrothermal atomic absorption spectrometry [11], flame atomic stripping voltammetry [12], adsorptive stripping voltammetry [13], plasma emission spectrometry [14], high performance liquid chromatography, separation [15] and membrane electrodes [16–24] have been listed for the detection of $Co^{+2}$. In spite of the previously published techniques having high sensitivity, some tedious complications were found in their utilities. Voltammetry is a cheap method, but with a very tedious procedure. Potentiometric detection depends on the membrane sensor, is very simple and introduces many excellent properties, like easy sample preparation, fast response, highly selective, wide linear concentration range, simple apparatus with very low detection limit, completed in viscous, colored and/or turbid solutions and less expensive [25]. However, most of the listed sensors suffering from calcium interference and have a very narrow concentration range.

New specific ligands related to heterodiazo dyes that form strong, stable complexes with $Co^{+2}$ ions were prepared to detect it in different samples by new, very sensitive and selective spectrophotometric detection [5–9].

The reagent 2-(5-Bromo-2-pyridylazo)-5-[N-n-propyl-N-(3-sulfopropyl) amino] aniline (BrPPSAA), Figure 1 has not only good sensitivity but also a very good selectivity behavior [9]. So, we decided to use its analytical usefulness and utility to construct $Co^{+2}$ membrane sensor.

**Figure 1.** Structure of the reagent [BrPPSAA].

The present work describes the construction and evaluation of the newly cobalt (II) membrane sensor. The active constituents in polyvinyl chloride (PVC) matrix selective sensors are the $Co^{+2}$ with the cited reagent [BrPPSAA] ion associate complex. The sensor is successfully applied for the estimation of the concentration of cobalt (II) ions in food products and pharmaceutical formulations.

## 2. Materials and Methods

### 2.1. Sample Products, Materials and Reagents

Chlorides of cobalt, zinc, nickel, cadmium, sodium, aluminum and/or calcium, hydrogen peroxide, ammonium and sodium hydroxides. Polyvinyl chloride (PVC) and TEHP (tri-(2-Ethylhexyl) phosphate) were Aldrich products. Hydrochloric, hydrofluoric and sulfuric acids, tetrahydrofuran, TBP (tributylphosphate) and methanol from Merck (Germany). Pharmaceutical samples containing cobalt; (Neurobion and Basiton fortes) and food products containing cobalt (spinach leaves, garlic root, red onion root, cinnamon, flour and powder milk) were obtained from the local agricultural farms and markets in Egypt and Saudi Arabia.

### 2.2. Preparation of the Cited Reagent

The cited reagent (BrPPSAA) was prepared and purified by means of crystallization by column chromatography as reported before [9].

### 2.3. Preparation of Stock Solutions

In fact, metal salts were weighed and dissolved in water to prepare 0.1 molar solutions, which were used as stock solutions in the present study. Solutions of $10^{-8}$–$10^{-2}$ M were prepared by dilution.

Cobalt chloride standard solutions used in determination of $Co^{+2}$ ions in pharmaceuticals and food products were accurately weighed, then dissolved in 0.01 M NaCl and diluted.

### 2.4. Sample Preparation for the Estimation of Cobalt Ions

The required solutions for potentiometric measurements were prepared as follows: a content of food products (spinach leaves, garlic root, red onion root, cinnamon, flour and powder milk) were selected for analysis. For the analysis of $Co^{+2}$ ion of garlic and red onion roots, samples were cut, washed and then heated at 120 °C for 2 h. Weight accurately 10 g, transferred (430 and 165 mg of them, respectively) into a crucible, then heated at 600 °C for 4 h for ashing, after completing the ashing the samples were cooled to the ambient temperature, 5 mL of 0.1 M HCl was added for dissolving the resulting residues, transferred into a 50 mL calibrated flask and diluted with bidistilled $H_2O$ to the mark (50 mL). For cinnamon, powder milk and flour, ten grams of sample was weighed accurately and put (600, 650 and 170 mg of them, respectively) into a quartz crucible.

Of 2 M nitric acid 10 mL was added, evaporated to dryness and 30% $H_2O_2$ was added dropwise until a clear solution and then evaporated. Adding bidistilled water and heating continuously to remove $H_2O_2$. The yield was cooled and diluted with bidistilled water. With respect to the analysis of spinach leaves, 5 g sample was weighed accurately, then taking 450 mg, the same procedure as mentioned above was followed.

For pharmaceutical formulations containing cobalt a content of Neurobion and Basiton fortes (500 and 550 mg of them, respectively) was transferred into a conical flask, adding 10 mL of 30% $H_2O_2$, left to stand until dissolving. Adding 1 mL of 1 M $H_2SO_4$, heating until $H_2O_2$ analyzed. This step was repeated six times. After mineralization adding 25 mL water and 10 mL ammonia solution, left to stand for an hour. After that, filter the solution quantitatively and diluted with bidistilled water to the mark (50 mL). Taking 10 mL volumes of each solution for $Co^{+2}$ ion determination at the optimum conditions of the developed method. the obtained data are summarized in Table 3.

### 2.5. Construction of the Membrane Sensor

The sensor membrane construction was introduced as described before [26]. It includes a column electrode of Teflon exchangeable and a body full of a membrane liquid phase + Ag/AgCl an internal reference electrode.

The polyvinyl chloride (PVC), the complex and plasticizer were fine powdered, then adding tetra hydro furan as a volatile solvent. A suitable diameter disk was cut and glued to the flat end of polyvinyl chloride (PVC) tubing with tetrahydrofuran (THF). The body of the sensor was filled with 0.001 molar solution of the specific cobalt sensor. The sensor was adapted by immersing for 24 h in 0.01 molar $Co^{+2}$ solution and stored for the rest of the period in the same solution.

### 2.6. Active Component of Liquid-Membrane Layer

The reagent 2-(5-Bromo-2-pyridylazo)-5-[N-n-propyl-N-(3-sulfopropyl) amino] aniline (BrPPSAA) is the active membrane component. It forms a cobalt complex with maximum absorption at 602 nm at the optimum pH range of 3–4.5, the molar absorptivity of this complex is $8.8 \times 10^4$ L $Mol^{-1}$ $cm^{-1}$, which is considered a high value. In addition, it is a water-soluble reagent, so no need here for solvent extraction and it is available at convenient costs [9].

### 2.7. The Potential Layer Preparation

An accurate weight 0.01 g active component (Co (BrPPSAA)) with a mixture of 0.55 g TEHP, 0.25 g PVC and 0.34 g TBP were mixed to prepare the membrane sensor's layer. A Teflon sensor with an electrode of Ag/AgCl was filled with the freshly prepared mixture, then transforming to gel by heating at a temperature 375 K for 30 min. After cooling, the electrode was soaked for two hours in $10^{-3}$ M cobalt ion solution.

### 2.8. EM F Measurements

An Orion 90-00-01 solution containing 1.5 M potassium nitrate, 0.55 M potassium chloride, 0.05 M sodium chloride and 40% formaldehyde one ml was used to fill the stable reference electrode's bridge. An Orion 90-02 reference electrode was used with a mechanical stirrer to give an accuracy of 0.1 mV at room temperature for measuring the EMF of the cobalt sensor system.

## 3. Results

The basic analytical parameters of our constructed cobalt membrane electrode were studied to determine its value in analytical applications. The selectivity behavior, limit of detection, the characteristic slope, dynamic time of response and the effect of pH on the sensor's potential were investigated.

### 3.1. Calibration Curves

Figure 2, the cobalt sensor's calibration curves presented a good linearity detected in $Co^{+2}$ ions in the concentration range of $10^{-8}$–$10^{-2}$ molar solutions.

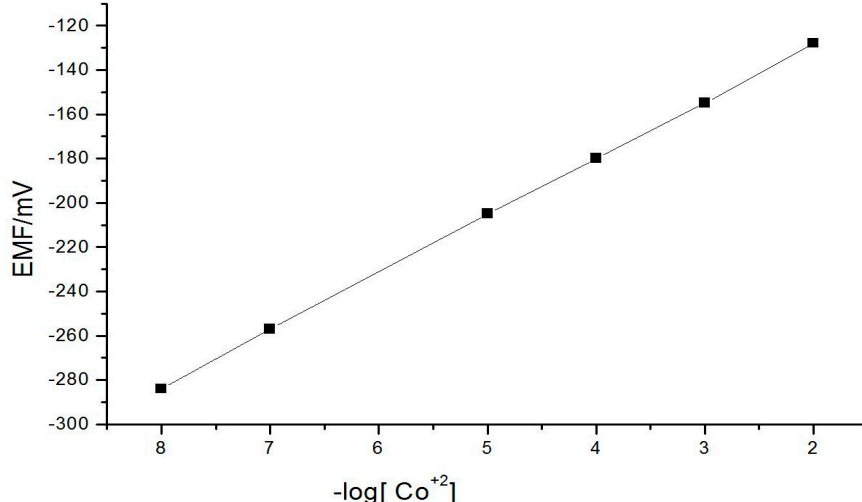

**Figure 2.** Calibration curve of the Co (II) membrane sensor in the concentration range $10^{-8}$–$10^{-2}$ M.

The cobalt sensor's characteristic slope was 56.66 mV, the limit of detection was $2.7 \times 10^{-8}$ molar and the measuring linear concentration range was $3.4 \times 10^{-8}$–$2.4 \times 10^{-2}$ molar. Table 1 presented the analytical characteristic parameters of the fabricated cobalt sensor.

**Table 1.** Analytical characteristics parameters of the proposed $Co^{+2}$ sensor matrix (reference electrode Ag/AgCl) membrane sensor preparation.

| | |
|---|---|
| Slope of the characteristics/mV | 56.66 |
| Intercept/mV | $-48.5 + 0.3$ |
| Limit of detection/mol dm$^{-3}$ | $2.7 \times 10^{-8}$ |
| Measuring range/mol dm$^{-3}$ | $3.4 \times 10^{-8}$–$2.4 \times 10^{-2}$ |
| Response time/s | 10 |
| Lifetime/d | 210 |
| pH range | 3.5–8.5 |

### 3.2. Selectivity Coefficient Sensor's Measurements

The selectivity coefficients of the $Co^{+2}$ membrane sensor with reference to interfering ions were studied by the separate solution or by the MP (matched potential) methods reported before [26] using the equations:

$$\text{llog } K_{ij}^{pot} = E_2 - E_1/S - (Z_i/Z_j - 1) \ \log \ a_i, \ K_{pot} \ Co/M = \frac{ai}{ai\frac{zi}{zj}}$$

By using the separate solution method, at the EMF value of $Co^{+2}$ ions with the concentration 0.001 M and the potential $-160$ mV. With respect to the matched potential method, the equation is: $K_{pot} \ Co/M = \frac{ai}{ai\frac{zi}{zj}}$.

The obtained data are reported in Table 2.

**Table 2.** The values of selectivity coefficients (K) of $Co^{+2}$ electrode matrix (reference electrode Ag/AgCl) membrane sensor preparation.

| | Separate Solution Method (SSM) | | Matched Potential Method |
|---|---|---|---|
| K | Ei = Ej | ai = aj | MPM |
| $CoCl_2$ | 0.326 + 0.0210 | 0.366 + 0.012 | 0.345 + 0.0180 |
| $CdCl_2$ | 0.240 + 0.0048 | 0.312 + 0.032 | 0.287 + 0.0110 |
| $NiCl_2$ | 0.066+ 0.0018 | 0.151 + 0.003 | 0.256 + 0.0230 |
| $AlCl_3$ | 0.046 + 0.0130 | 0.054 + 0.002 | 0.004 + 0.0012 |
| $CaCl_2$ | 0.245 + 0.0050 | 0.321 + 0.040 | 0.308 + 0.0060 |
| $CuCl_2$ | 0.014 + 0.0003 | 0.066 + 0.002 | 0.016 + 0.0003 |

### 3.3. Response Time of the Proposed Cobalt Electrode

The constructed membrane ion selective electrode response time is very important for analytical applications. After injecting the standard concentrated solution, adding water (1:1) for dilution. Solutions used for the determination of the response time for the tested sensor have these conditions: $c_1$:$c_2$ = 1:100 and $v_1$:$v_2$ = 1: 20, where $c_1$ is the concentration of the sample, $c_2$, the standard concentration, $v1$ is the sample volume and $v_2$ is the volume of standard. The obtained data are presented in Figure 3. The response of the sensor was reproducible after 10 s of adding cobalt. The timer was started at the instant of injection of the concentrated sample, the fast and stabilized reading of potential reflecting the time required for completing the titration. As can be seen from Figure 3, the electrode reached its equilibrium response in a very short time (10 s) over the whole concentration range.

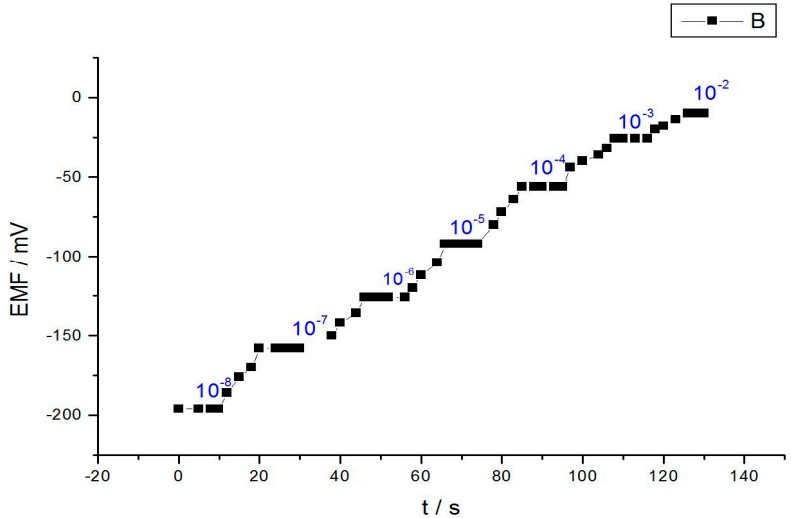

**Figure 3.** Response time of cobalt sensor ions concentration $10^{-3}$ M.

### 3.4. Dependence of The Electrode Potential on the pH

The dependence of the electrode potential on the pH was studied by measuring the potential, according to the chemical character of cobalt salts. Hydrochloric acid or drops of sodium hydroxide were added to the 0.001 M cobalt ions concentration sample under investigation. After each addition of the acid or base the pH was registered, the ratio of the electromotive force (EMF) of the cobalt sensor system/reference electrode was read after the sensor's response stabilized. The effect of pH on the EMF was introduced in Figure 4. Below and above this pH range (3.5–8.5), at higher pH values, the potential decreases (−175 at pH 9, −178 at pH 9.5 and −184 at pH 10) might be attributed to the non-completing complex formation or the hydrolysis of $Co^{+2}$ ions. At lower pH values, potential increases

$(-135$ at pH 2.4, $-144$ at pH 2.7 and $-150$ at pH 3) were due to the membrane responses to hydronium ions and Co (II) ions.

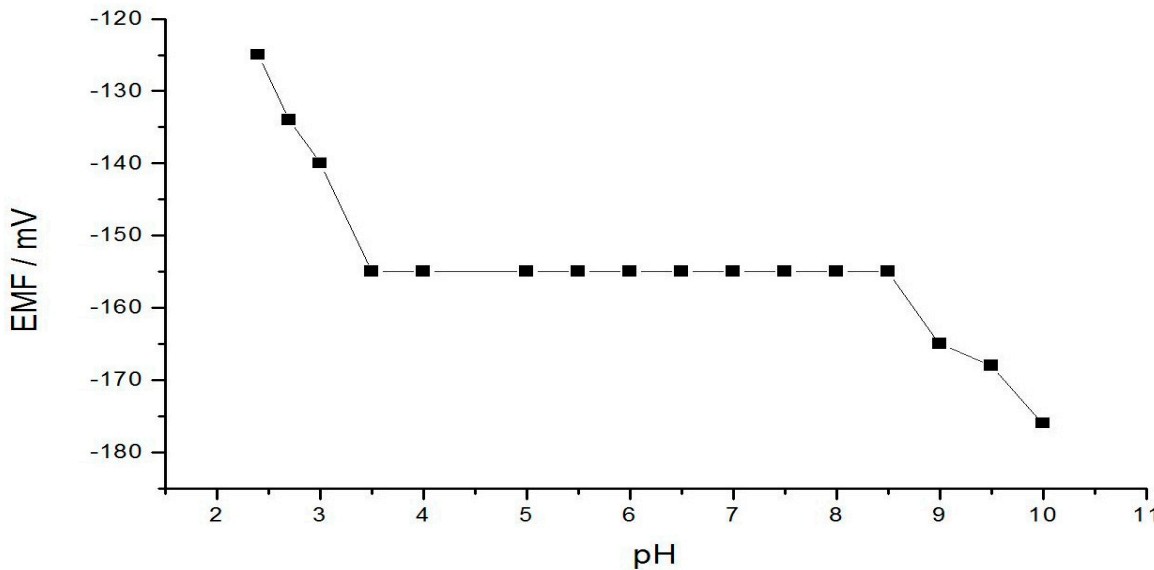

**Figure 4.** Effect of the sensor response on the pH in $10^{-3}$ M cobalt ions concentration.

### 3.5. Lifetime of the Cobalt Sensor

The lifetime of the tested cobalt sensor under investigation was studied by measuring the characteristic slopes of the sensor stored in dry air. The regular tests were completed once a week for 10 months, in freshly prepared solutions in a systematic way. Stable and reproducible signals were acquired during 7 months. It was observed that a slight decrease in the slope of the electrode by 1.0 mV decade-1 from 56.66 to 55.66 mV/decade and an increase in the detection limit. After the end of the time period, the slope of the electrode decreased gradually, whereas the detection limit was increased (from 51.23 to 47.15 5 mV per decade and $2.6 \times 10^{-7}$ to $4.2 \times 10^{-6}$ M, respectively). This probably arises from the leaching of the membrane components. Therefore, the lifetime of the electrode was about 7 months, according to the basis of the obtained data.

### 3.6. Determination of Cobalt in Food Products and Pharmaceutical Formulations Samples

The determination of $Co^{+2}$ ions in food products and pharmaceutical formulations samples was studied using the formed membrane electrode to investigate its analytical usefulness. The standard additions and the calibration curve methods were used, the determined results and their statistical treatment are introduced at Table 3.

**Table 3.** Determination of $Co^{+2}$ in food and pharmaceutical samples using matrix (reference electrode Ag/AgCl) membrane preparation.

| Sample (Active Principal) | Calibration Curve Method | | | | Standard Addition in the Sample with Dilution | | | |
|---|---|---|---|---|---|---|---|---|
| | Sample Data mg/Kg | $Co^{+2}$ Found mg/Kg | Relative Error % | V % | Sample Data mg/Kg | $Co^{+2}$ Found mg/Kg | Relative Error % | V % |
| Garlic Root | 430 | 430.65 | 0.14 | 0.08 | 430 | 432.75 | 0.60 | 0.15 |
| Flour | 170 | 171.45 | 0.97 | 0.05 | 170 | 172.86 | 1.91 | 0.12 |
| Powder Milk | 650 | 651.55 | 0.21 | 0.23 | 650 | 652.89 | 0.39 | 0.25 |
| Spinach Leaves | 450 | 451.86 | 0.34 | 0.12 | 450 | 452.95 | 0.54 | 0.16 |

<div align="center">Table 3. <i>Cont.</i></div>

| Sample (Active Principal) | Calibration Curve Method | | | | Standard Addition in the Sample with Dilution | | | |
|---|---|---|---|---|---|---|---|---|
| | Sample Data mg/Kg | Co$^{+2}$ Found mg/Kg | Relative Error % | V % | Sample Data mg/Kg | Co$^{+2}$ Found mg/Kg | Relative Error % | V % |
| Red Onion Root | 165 | 166.76 | 1.00 | 0.26 | 165 | 167.54 | 1.45 | 0.16 |
| Cimamon | 600 | 601.85 | 0.46 | 0.35 | 600 | 602.75 | 0.69 | 0.28 |
| Neurobion forte [a] | 500 | 501.65 | 0.33 | 0.11 | 500 | 502.15 | 0.43 | 0.51 |
| Basiton forte [b] | 550 | 551.75 | 0.35 | 0.13 | 550 | 502.65 | 0.53 | 0.26 |

The averages of (five) estimations. [a] The Procter and Gamble Company. [b] Piramal Health. V = $\frac{\delta n - 1}{x} \times 100\%$—The values reported in the "Sample Data" those give the accurate results in five measurements.

## 4. Discussion

Nernstian slope is a very important factor to detect electrodes and was suitably used in the analysis. It was calculated from the slope curve between the log concentrations of standard solution (M) with the potential measured in EMF/mV as shown in Figure 2. The ideal value of Nernstian slope was 59.1/n (mV/decade), where n is the valency [27]. This means that for the Co$^{+2}$ ion selective electrode with the n value = 2 it was 29.6 mV/decade. The Nernstian factor value in this study was 28.33 mV/decade, which means any increase in the concentration of $10^{-1}$ M test solutions, there is a potential change of 28.33 mV/decade. This value indicates that the Co$^{+2}$ ion selective electrode was still feasible for use in the analysis of Co$^{+2}$, because the allowed value of Nernstian slope was 28.33 mV.

In this study, a variety of different cations were investigated as interfering ions. The selectivity coefficient values given in Table 2 revealed that the proposed cobalt ion membrane sensor was highly selective towards cobalt (II), satisfactorily in the presence of Cd, Ni, Al, Ca and Cu. As it was obvious from the resulting data, none of the tested interfering cations had a significant influence on the potentiometric responses of the electrode towards cobalt ions. Clearly, for all diverse ions used, the selectivity coefficients were smaller, indicating they would not significantly disturb the Co$^{+2}$ ion-selective electrode function. The surprisingly high selectivity of the membrane electrode for cobalt ions over other cations used, most probably arises from the strong tendency of the carrier molecules for cobalt ions.

The proposed sensor was successfully applied to the direct determination of Co$^{+2}$ ions in food products. Furthermore, the applicability of the proposed sensor was also tested to determine cobalt in pharmaceutical formulations without prior extraction as shown in Table 3. The results obtained by the proposed sensor were quite quantitative, precise and accurate.

Additionally, as can be seen from the obtained results (Table 3) that the standard additions and the calibration curve methods were applied. The data analysis shown that the method of calibration curve was preferred in the determination of Co$^{+2}$ while the standard additions method was less recommended. Therefore, the error was no bigger than 1% and 1.91% in the two methods, respectively, this is due to the reproducibility of the method.

The obtained data by the constructed Co$^{+2}$ electrodes were analyzed and compared with the other different cited electrodes. Table 4 introduced a comparison between some of the characteristics of the quantitative determination of Co$^{+2}$ ions using different electrodes reported in the literature. This comparison was made to indicate, whether the proposed electrode gives reliable results and be accepted for Co$^{+2}$ ions determination in food products and pharmaceutical samples. It can be shown from Table 4 that the proposed electrode exhibited comparable linear concentration range ($3.4 \times 10^{-8}$–$2.4 \times 10^{-2}$ M), which is more valuable than the other published Co$^{+2}$ electrodes [16–24]. It has a long shelf life (210 days) compared to the other cited electrodes, all sensors had low limits of detection, the lowest of them was that stated in this work ($2.7 \times 10^{-8}$). Further, the proposed electrode has

many advantages as compared with others, it is easy to fabricate and it is plainly affordable. Therefore, it can be trusted to say that our sensor is applicable in all senses with other electrodes for the determination of $Co^{+2}$ ions.

**Table 4.** Comparison of important parameters of Co (BrPPSAA) with some recently cited sensors for $Co^{+2}$ ions estimation.

| Ref. | Slope (mV) | Linear Range (M) | Lifetime | Detection Limit (M) | pH Range |
|---|---|---|---|---|---|
| this work data | 56.66 | $3.4 \times 10^{-8}$–$2.4 \times 10^{-2}$ | 7 months | $2.7 \times 10^{-8}$ | 3.5–8.5 |
| 16 | 30.5 | $1.9 \times 10^{-5}$–$1.0 \times 10^{-1}$ | 4 months | ——— | 1.9–5.8 |
| 17 | $29.5 \pm 0.2$ | $6.3 \times 10^{-7}$–$1.0 \times 10^{-1}$ | ——— | $3.9 \times 10^{-7}$ | 2.5–6.5 |
| 18 | ——— | $1.0 \times 10^{-6}$–$1.0 \times 10^{-1}$ | 2 months | $5.0 \times 10^{-7}$ | 4.0–9.5 |
| 19 | $30 \pm 0.2$ | $7.9 \times 10^{-8}$–$1.0 \times 10^{-1}$ | 5 months | $5.0 \times 10^{-8}$ | 2.0–9.0 |
| 20 | $65.8 \pm 1.0$ | $1.0 \times 10^{-5}$–$1.0 \times 10^{-1}$ | ——— | $6.0 \times 10^{-6}$ | 2.0–5.5 |
| 21 | ——— | $6.58 \times 10^{-7}$–$1.0 \times 10^{-1}$ | ——— | $6.82 \times 10^{-8}$ | 2.5–8.5 |
| 22 | $29.5 \pm 0.4$ | $2.5 \times 10^{-7}$–$1.0 \times 10^{-1}$ | 2 months | $6.1 \times 10^{-8}$ | 2.0–7.0 |
| 23 | $30.2 \pm 1.0$ | $1.1 \times 10^{-7}$–$1.0 \times 10^{-1}$ | ——— | ——— | 2.0–8.0 |
| 24 | $25.3 \pm 1.0$ | $1.0 \times 10^{-6}$–$1.0 \times 10^{-1}$ | 4 months | $3.5 \times 10^{-7}$ | 3.5–6.5 |

No interference was detected from the excipients found in the samples under test. The calibration curves showed a good linear response on a wide range of concentrations. Most of the methods perform a valuable recovery with respect to the known values and there is no significant variation for either accuracy or precision were introduced.

## 5. Conclusions

A fabricated cobalt membrane electrode was introduced. The proposed electrode was characterized by excellent analytical parameters: for the Nernstian slope, short response time and relatively long lifetime. The analytical properties of the investigated electrode are shown in Tables 1 and 2.

The formed electrode was used for $Co^{+2}$ ion determination in food and pharmaceutical samples that are commonly used. The standard additions and the calibration curve methods were applied. The data analysis shown that the method of calibration curve was preferred in the determination of $Co^{+2}$ while the standard additions method was less recommended. Therefore, the error was no bigger than 2% due to the reproducibility of the method. The method of fabricated electrode was found as precise and accurate as compared to other reported techniques, which is widely used in their determination in food and pharmaceutical samples (Table 4).

Generally, the quality of the results was very good, which is attributed to the importance of the selected applications using the constructed electrode. The time taken in the analyses was studied without any effect on the accuracy, precision and reproducibility of the results.

**Author Contributions:** Software, M.E.-S.; formal analysis, M.E.-S.; resources, M.E.-S.; writing—original draft preparation, S.K.; writing—review and editing, S.K.; supervision, S.K. All authors have read and agreed to the published version of the manuscript.

**Funding:** This research received no external funding.

**Institutional Review Board Statement:** Not applicable.

**Informed Consent Statement:** Not applicable.

**Data Availability Statement:** The data that support the findings of this study are available on request from the corresponding author.

**Acknowledgments:** This work was carried out using the facilities and materials in Taif University Researches Supporting Project number (TURSP-2020/139), Taif University, Taif, Saudi Arabia.

**Conflicts of Interest:** The authors declare no conflict of interest.

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
