# Peer review of "Construction of a Highly Selective Membrane Sensor for the Determination of Cobalt (II) Ions"

_chemosensors, doi:10.3390/chemosensors9050086_

Round 1
Reviewer 1 Report
Correspondence reference: chemosensors-1155940
“Construction of a Highly Selective Membrane Sensor for the Determination of Cobalt (II) Ions”
Dear Editor,
The manuscript, “Construction of a Highly Selective Membrane Sensor for the Determination of Cobalt (II) Ions” describes a potentiometric sensor based on ion-selective electrode for cobalt ions detection in food products and pharmaceutical formulations. The results are interesting but, the key-part of the paper, "Results and Discussion" needs to be more complete with some investigations. I recommend rejection this manuscript version with special attention for:
- Please, add a picture of ISE.
- The paper is scientifically correct, but it is not interesting for the sensors community. It is very generic and naive. This paper presents the information at the level of a student who is a very new in the research area.
- This text and corresponding illustrations might be appropriate as an introduction to a student's MS thesis (not even for a PhD thesis).
- Professional researchers working in the sensors area will not find in this paper any information justifying reading the paper.
- Overall, the paper is certainly not at the scientific level needed for the article and it cannot be recommended for publication.
Author Response
My Current address
Dr. S. Khalil Mohamed
Food Nutrition Sci. Dept,
Faculty of Science,
Taif University
Taif 21974,
P.O. Box 888,
Kingdom of Saudi Arabia
Fax: + 96614915684
Tel: 00966543372745
E-mail: S_Khalil_99@Yahoo.co.uk
& sabry@Tu.edu.sa
March., 28, 2021
Editor- in-Chief
Chemosensors
Dear Sir,
Thank you for your e-mail dated 24 March., 2021 concerning the manuscript of ID reference : Chemosensors-1155940, entitled: “Construction of a Highly Selective Membrane Sensor for the Determination of Cobalt (II) Ions ”
The reviewer comments were considered, the paper has been revised and re-arranged again according to the Journal style. The amendments and modifications done are in the following list:-
- At page 2 line 51 [ BrPPSCHN ] was corrected to [ BrPPSAA]
- Section 2.4. Sample preparation was revised, the proper concentrations and the amounts in each case were reported. Also, the format of all amounts were adjusted in all manuscript.
- At page 3 lines 112-114 the sentence was rephrased.
- Figure 1 was cited in the text at page 2 line 46.
- In the response time section the word mercury was corrected by cobalt and the word reproducable was corrected by reproducible at page 4 line 153.
- Table 4 was revised to show the missing parts.
- The first paragraph of the discussion section which is repeated in the results section was canceled from the results section.
- References numbers 18 and 22 which are dont deals with cobalt determination were canceled from Table 4 and References List.
- References numbers 28 , 29 were replaced by more appropriate articles [ 5-9 ].
- All References in the text are re-ordered and marked with red color.
- IN Figure 2 the slope was calculated from the top and bottom points which represent the fitting linear line, the caption of the figure letters was revised in the proper order, pa in the X-axis refers to " - log [ metal ion concentration ] " and the concentration of cobalt and its interfering ions is 10-3 M solutions.
- Selectivity Section was discussed at page 8 lines 219-223.
- The concentration of cobalt ions at which the pH evaluation experiment was done is 0.001 M and it was introduced at page 4 line 161.
- In Figure 4 there was a straight line in the curve at pH range [ 3.5 - 8.5 ]and not one point.
- In the section Lifetime of the cobalt sensor we show some results that support the results at page 4 lines 170-173.
- The section Determination of cobalt in food products and pharmaceutical formulations was discussed at page 8 lines 222-230.
Thanking you in anticipation, please accept my best regards.
Yours Sincerely,
Prof. Dr. Sabry Khalil
Reviewer 2 Report
Comments to the author:
In this manuscript, the authors presented the fabrication of a novel membrane sensor for the highly selective sensing of cobalt (II) ions. Although the use of the selective receptor 2-(5-bromo-2-pyridylazo)-5-[N-n-propyl-N-(3-sulfopropyl) amino] 9 aniline (BrPPSAA) is not completely new, the authors have done a great job here because they have analyzed all the parameters important in a sensor, specially they have proven the selectivity of this receptor. Besides they have reported the sensitivity, limit of detection, response time, shelf stability, pH dependance and also they have assessed their sensor in real food samples. Overall, I would recommend the publication of this contribution but some issues should be addressed first.
The following are some questions and suggestions for improving their work:
Major issues:
- Figure 2 is not clear. How this experiment was done? It is the calibration curve of Co2+ with/without the presence of interfering cations? If so, which are the concentrations? Did you use always the same concentration of interfering cations or did you use the same as Co2+? For this kind of experiments usually the concentration of interfering molecules is fixed to be 10 or 100 times the one of the analyte of interest.
- In the same figure, the linear fitting should be shown. Also in the caption of the figure letters should be shown in the proper order. Also it is not clear what is pa in the X axis.
- Selectivity section should be discussed. Only the table and figure 2 by themselves are not enough. It is not clear for the reader if selectivity is achieved or not. Please discuss this section.
- The pH evaluation experiment at what concentration of Co2+ was done? In figure 4 the authors have written a concentration range but they have only showed one point per pH.
- In the section of Lifetime of the Cobalt Sensor the authors should show some results that support their claims.
- As for the selectivity section, the section ‘Determination of Cobalt in Food Products and Pharmaceutical Formulations Samples’ should be discussed. Only the table is not enough. It is not clear the difference between calibration curve and standard addition columns. What have the authors done here?
Minor issues:
- Page 2, line 51, the authors have written [BrPPSCHN] instead of [BrPPSAA].
- In the section: 2.4. Sample Preparation for the Estimation of Cobalt ions there are many information missing. The authors should report proper concentration values and protocols. For instance: i) 5 ml of diluted HCl, ii) Ten ml concentrated nitric acid, iii) diluted with bidistilled H2O. Please report the proper concentrations and amounts in each case. Also be consistent with the format you use, either 5 mL or five mL, but the same format should be use in all manuscript.
- Although section 2.5. Construction of the Membrane Sensor has been already reported the exact protocol should also be in this manuscript.
- Page 3, line 110. What the authors want to express here? ‘An accurate weight 0.01g active component [ Co ( BrPPSAA ) ] with a mixture of, 110 0.55 g TEHP, 0.25 g PVC and 0.34 g TBP and perform the membrane sensor’s layer.’ This sentence should be rephrased.
- Figure 1 is not cited in the text.
- In the response time section the authors have written mercury instead of cobalt and reproducable instead od reproducible.
- Some of the text in Table 4 is cut. Please revise.
- First paragraph of the discussion section is exactly the same as in the results section. Please revise.
Author Response

(The authors gave the same response as above.)

Round 2
Reviewer 1 Report
Correspondence reference: chemosensors-1155940-v2
“Construction of a Highly Selective Membrane Sensor for the Determination of Cobalt (II) Ions”
Dear Editor,
The manuscript, “Construction of a Highly Selective Membrane Sensor for the Determination of Cobalt (II) Ions” describes a potentiometric sensor based on ion-selective electrode for cobalt ions detection in food products and pharmaceutical formulations. The author made systematic revisions based on the reviewers' comments, and I recommend the publication of the manuscript in the current form.
Author Response
As mentioned before, Reviewer1 Recommended the publication of the manuscript in the current form.
My Current address
Prof. Dr. Sabry Khalil
Food Nutrition Sci. Dept,
College of Science,
Taif University
Taif 21944,
P.O. Box 11099,
Kingdom of Saudi Arabia
Fax: + 96614915684
Tel: 00966543372745
E-mail: S_Khalil_99@Yahoo.co.uk
& sabry@Tu.edu.sa
April,10, 2021
Editor- in-Chief
Chemosensors
Dear Sir,
Thank you for your e-mail dating April, 8, 2021 concerning the manuscript of ID reference : Chemosensors-1155940, entitled: “Construction of a Highly Selective Membrane Sensor for the Determination of Cobalt (II) Ions ”
The reviewer comments were considered, the paper has been revised for the second time. The amendments and modifications done are in the following list:-
- Section 2.5 " Construction of the membrane sensor " The exact protocol was stated in the manuscript at page 3 lines 103-109.
- In the section of Lifetime of the cobalt sensor we show some results that support the claims at page 4 lines 178-187.
- The Selectivity of cobalt sensor section was re-discussed at page 8 lines 230-238.
- The Determination of cobalt sensor in food products and pharmaceutical formulation samples section was re-discussed at page 8 & 9 lines 239-248.
- Figure 2 was re-plotted to show the linear fitting.
- All changes in the text are marked with red color.
Thanking you in anticipation, please accept my best regards.
Yours Sincerely,
Prof. Dr. Sabry Khalil

Reviewer 2 Report
The authors did not reply to all my concerns:
- Although section 2.5. Construction of the Membrane Sensor has been already reported the exact protocol should also be in this manuscript.
- In the section of Lifetime of the Cobalt Sensor the authors should show some results that support their claims.
- The selectivity and Determination of Cobalt in Food Products and Pharmaceutical Formulations Samples sections are poorly discussed.
- In figure 2, the linear fitting should be shown. Also in the caption of the figure letters should be shown in the proper order.
Author Response
My Current address
Prof. Dr. Sabry Khalil
Food Nutrition Sci. Dept,
College of Science,
Taif University
Taif 21944,
P.O. Box 11099,
Kingdom of Saudi Arabia
Fax: + 96614915684
Tel: 00966543372745
E-mail: S_Khalil_99@Yahoo.co.uk
& sabry@Tu.edu.sa
April,10, 2021
Editor- in-Chief
Chemosensors
Dear Sir,
Thank you for your e-mail dating April, 8, 2021 concerning the manuscript of ID reference : Chemosensors-1155940, entitled: “Construction of a Highly Selective Membrane Sensor for the Determination of Cobalt (II) Ions ”
The reviewer comments were considered, the paper has been revised for the second time. The amendments and modifications done are in the following list:-
- Section 2.5 " Construction of the membrane sensor " The exact protocol was stated in the manuscript at page 3 lines 103-109.
- In the section of Lifetime of the cobalt sensor we show some results that support the claims at page 4 lines 178-187.
- The Selectivity of cobalt sensor section was re-discussed at page 8 lines 230-238.
- The Determination of cobalt sensor in food products and pharmaceutical formulation samples section was re-discussed at page 8 & 9 lines 239-248.
- Figure 2 was re-plotted to show the linear fitting.
- All changes in the text are marked with red color.
Thanking you in anticipation, please accept my best regards.
Yours Sincerely,
Prof. Dr. Sabry Khalil